# Surgical De-Escalation for Re-Excision in Patients with a Margin Less Than 2 mm and a Diagnosis of DCIS

**DOI:** 10.3390/cancers16040743

**Published:** 2024-02-10

**Authors:** Gianluca Vanni, Marco Pellicciaro, Nicola Di Lorenzo, Rosaria Barbarino, Marco Materazzo, Federico Tacconi, Andrea Squeri, Rolando Maria D’Angelillo, Massimiliano Berretta, Oreste Claudio Buonomo

**Affiliations:** 1Breast Unit Policlinico Tor Vergata, Department of Surgical Science, Tor Vergata University, Viale Oxford 81, 00133 Rome, Italy; gianluca.vanni@ptvonline.it (G.V.); mrcmaterazzo@gmail.com (M.M.); oreste.buonomo@ptvonline.it (O.C.B.); 2Ph.D. Program in Applied Medical-Surgical Sciences, Department of Surgical Science, Tor Vergata University, 00133 Rome, Italy; 3Department of Surgical Sciences, Tor Vergata University, 00133 Rome, Italy; nicola.di.lorenzo@uniroma2.it; 4Radiotherapy, Department of Oncoematology, Policlinico Tor Vergata, 00133 Rome, Italy; rosaria.barbarino@ptvonline.it (R.B.); rolandomaria.dangelillo@ptvonline.it (R.M.D.); 5Department of Surgical Sciences, Unit of Thoracic Surgery, Tor Vergata University, 00133 Rome, Italy; federico.tacconi@ptvonline.it; 6School of Specialization in Medical Oncology Unit, Department of Human Pathology “G. Barresi”, University of Messina, 98100 Messina, Italy; andrea.squeri@studenti.unime.it; 7Department of Clinical and Experimental Medicine, University of Messina, 98100 Messina, Italy; berrettama@gmail.com

**Keywords:** DCIS, ductal carcinoma in situ, omitting margin re-excision, surgical de-escalation, locoregional recurrence

## Abstract

**Simple Summary:**

Ductal carcinoma in situ is a malignant cell proliferation confined to basement membrane. Current consensus guidelines recommend an optimal margin width of 2 mm and re-excision for closer margin is debated and it is sent back to clinical judgment. Our retrospective study evaluating 197 patients aim to investigate the importance of surgical margin and locoregional recurrence in patients with diagnosis of DCIS and treated with conservative breast surgery. We found no correlation between margins and loco-regional recurrence, and re-excision should be avoided in patients with focally positive margin and no evidence of the disease at post-surgical imaging.

**Abstract:**

The current surgical guidelines recommend an optimal margin width of 2 mm for the management of patients diagnosed with ductal carcinoma in situ (DCIS). However, there are still many controversies regarding re-excision when the optimal margin criteria are not met in the first resection. The purpose of this study is to understand the importance of surgical margin width, re-excision, and treatments to avoid additional surgery on locoregional recurrence (LRR). The study is retrospective and analyzed surgical margins, adjuvant treatments, re-excision, and LRR in patients with DCIS who underwent breast-conserving surgery (BCS). A total of 197 patients were enrolled. Re-operation for a close margin rate was 13.5%, and the 3-year recurrence was 7.6%. No difference in the LRR was reported among the patients subjected to BCS regardless of the margin width (*p* = 0.295). The recurrence rate according to margin status was not significant (*p* = 0.484). Approximately 36.9% (n: 79) patients had resection margins < 2 mm. A sub-analysis of patients with margins < 2 mm showed no difference in the recurrence between the patients treated with a second surgery and those treated with radiation (*p* = 0.091). The recurrence rate according to margin status in patients with margins < 2 mm was not significant (*p* = 0.161). The margin was not a predictive factor of LRR *p* = 0.999. Surgical re-excision should be avoided in patients with a focally positive margin and no evidence of the disease at post-surgical imaging.

## 1. Introduction

Intraductal carcinoma, also referred to as ductal carcinoma in situ, is a non-invasive breast carcinoma confined to the mammary ductal–lobular system with inherent but not mandatory tendency to progression to invasive carcinoma [1].

Surgery remains the first choice for patients with a DCIS diagnosis at core needle biopsy due to the inability to reliably exclude invasion; exceptions are made in some clinical trials and in frail patients with a severe comorbidity [2,3,4].

In the current practice, surgical management options are various and include breast-conserving surgery with or without adjuvant radiation therapy and mastectomy [3]. The surgical procedure selection is determined by the ratio between the lesion and the breasts’ size, pathologic findings, and the patients’ preference [3].

The ideal candidates for breast-conserving surgery are usually women with a small lesion, limited microcalcifications, and a monocentric lesion [5]. Otherwise, a destructive intervention, such as mastectomy, for non-invasive carcinoma with a tendency towards progression is strongly advised but not mandatory. Currently, the management of DCIS continues to be challenging. Minimally invasive surgery should be the first choice, followed by genetic testing to predict which tumor will progress to invasive carcinoma. One of the major challenges of breast-conserving surgery for DCIS is to achieve R0, in order to reduce future local recurrence [6]. Obtaining negative margins in DCIS could be difficult due to its presentations [6]. In situ lesions often present as vague masses, which cannot be clinically assessed by the physicians [7]. Nowadays, re-excision is the gold standard treatment whenever margins are positive or criteria are not met in DCIS [6]. The re-excision rate for positive or focally affected margins is approximately 30% due to underestimation of the true extent of the lesion in pre-operative imaging [8,9,10].

In almost all the analyses, the margin status seems to be strongly correlated with the incidence of local recurrence [11]. The current published guidelines recommend a margin threshold of 2 mm; however, the need for reoperation for closer margins is deferred to clinical judgment [12].

Due to unclear guidelines, the management of margins < 2 mm in DCIS patients could be different from country to country or even from breast clinic to breast clinic. The aim of our retrospective study is to evaluate the correlation between margins and locoregional recurrence (LRR) considering treatments adopted to our institution for close or involved margins.

## 2. Materials and Methods

Our retrospective study included all female patients with a DCIS diagnosis at final pathological examination, subjected to breast-conserving surgery, and evaluated between 2017 and 2021 in the breast unit of PTV (Policlinico Tor Vergata, Rome, Italy). The study was approved by the ethics committee of our hospital (approval number 72.23).

The patients’ demographics data were retrieved from clinical notes. Follow-ups of the patients were reported from surgical or oncological clinical notes. The locoregional recurrence (LRR) of DCIS was defined as disease recurrence in the ipsilateral breast, and it was categorized as invasive or in situ recurrence. The sample was divided into two groups according to the LRR: the locoregional recurrence group (LRR) and the no-recurrence group (NLR). The data on lesion characteristics were obtained from ultrasonography, mammography, or magnetic resonance. The BI-RADS (breast imaging reporting and data system of the American College of Radiology) was used to categorize the imaging findings. All the imaging was reviewed by at least two breast-dedicated radiologists. The lesion sites were described according to their position in the breast quadrants. Multifocal, multicentric, and unilaterality or bilaterality of breast cancer was reported. All the patients included had a histological preoperative diagnosis.

The analysis took into account the type of tumor, dimensions, grade, prognosis, and hormone receptor retrieved from the pathological examination.

Surgical axillary staging, according to whether a sentinel lymph node biopsy was performed or not, was reviewed from the surgical reports. Lymph node metastases reported in the final pathological examinations were categorized according to the current guidelines in isolated tumor cells (single tumor cell, or tumor-cell cluster < 0.2 mm), micro-metastasis (>200 cells or >0.2 mm, but <2.0 mm), or macro-metastasis (>2.0 mm).

A close margin was defined as the distance between the tumor and resection margin smaller than 2 mm. Surgical resection margins < 2 mm assed by pathological examination were included in the analysis. In the case of re-excision, the resection margin was considered only from the second procedure. Adjuvant treatments, such as hormone therapy and radiation therapy, were also considered in the analysis. Patients subjected to radiation therapy were stratified based on the type of treatment boost, standard, and number of fractions, daily dose, and total dose. The recurrence analysis included only re-excision within three months from the first surgery; all the patients treated with mastectomy as the second surgery were excluded.

### Statistical Analysis

All the data were reported into the prospective Excel database (Microsoft, Washington, DC, USA version 16.78, 2023). Based on the presence or absence of locoregional recurrence, the clinicopathological variables were compared between the groups using the T test for continuous variables. Fisher’s exact test was applied in cases of dichotomous variables, and the Monte Carlo test was used in cases of non-dichotomous variables. A multivariate logistic regression analysis was performed to identify the risk predictors of DCIS recurrence. A *p* value < 0.05 was considered to be statistically significant. The Kaplan–Meier curve was adopted to evaluate recurrence, and the log-rank test was used to assess statistically significant differences between the groups. A logistic regression statistical model was used to estimate the effect of factors on LLR. The Pearson test was adopted to appraise the correlation between variables. All the statistical analyses were performed in SPSS statistical package version 23.0 (SPSS Inc., Chicago, IL, USA).

## 3. Results

From January 2015 to December 2021, 255 female patients with a diagnosis of DCIS at final pathological examinations and a history of breast surgery were evaluated in the breast unit of PTV (Policlinico Tor Vergata, Rome). Approximately 16.1% (n: 41) underwent mastectomy and were therefore excluded from the study. Approximately 83.9% (n: 214) underwent breast-conserving surgery, with an overall median age of 53 years (range, 37–86 years). The median follow up was 5.5 years (range, 2.4–8.6 years). The 90-day reoperation rates were 13.5% due to incomplete tumor resection or close margins < 2 mm. Approximately 64.1% (n: 137) of the resection margins were >2 mm, while 35.9% (n: 77) were <2 mm.

All the patients subjected to secondary breast-conserving surgery underwent adjuvant radiation therapy. Approximately 13.7% (n: 35) did not need adjuvant radiation therapy. Out of 214 patients, 116 (54.2%) underwent adjuvant hormone therapy, and no cases of adjuvant chemotherapy were reported. Twenty-nine needed a second surgery, either a conservative one (n: 12; 41.4%) or mastectomy (n: 17; 58.6%), which were excluded for the LRR analysis.

Among 197 patients, two groups were identified: those who had local recurrence (LRR group; n: 15) and those who did not (NLR group; n: 182). The recurrence rate at 3 years’ follow-up was 7.6% (Figure 1a).

The median recurrence timing was 2.68 years (range, 1.3–6.1 years). Looking at the Van Nuys prognostic index, 42.6% (n: 84) had a low risk, 45.1% (n: 89) had an intermediate risk, and 1.1% (n: 2) had a high risk, while 11.6% (n: 22) did not have a calculable risk due to missing data.

The median age of the LRR group was 54.4 (range, 41–76 years) versus 57.3 (range, 37–86 years) of the NLR one, with no statistically significant difference (*p* = 0.381). The median follow-up was comparable in the two groups: 2.9 years (range, 2.6–7.8 years) in the LRR group versus 3.3 years (range, 1.1–8.1 years) (*p* = 0.247). The mean radiological maximum diameter of the lesion was 18.6 ± 9.1 mm in the LRR group versus 15.7 ± 9.3 mm (*p* = 0.242). The age score did not show any significant difference between the groups with a *p* = 0.692 (Table 1).

The tumor sites within different breast quadrants were significantly different between the LRR and NLR group, *p* = 0.007 (Table 1). The score dimension of the DCIS and cases of multifocal and/or multicentric lesions of the two groups had no significant differences (Table 1). Out of 15 cases of recurrence, 4 cases (26.7%) presented invasive cancer, and no cases of metastasis were reported with our follow-up.

The radiological findings were comparable in the two populations (Table 1). The histopathological features of the two groups were similar (Table 2).

Approximately 33.3% (n: 5) of the patients in the LRR group were also subjected to sentinel lymph node biopsy compared to 29.1% (n: 53) in the NLR, *p*-value: 0.772. No cases of lymph node metastasis or micro-metastasis were reported in the two groups.

The re-operation rate due to positive or <2 mm margins was 6.6% (n: 12) in the NLR group and none in the LRR group, *p* = 0.605.

The grading of in situ carcinoma was comparable between the groups. In the LRR group, 46.7% presented low-grade versus 31.3% in the NLR group; there were no cases of intermediate grade in the LRR group compared to 24.4% in the NLR one and 53.3% vs. 44.4% high-grade in the LRR and NLR groups, respectively, *p* value 0.056 (Table 2). Comedonecrosis at final pathological examination was present in 49.4% (n: 81) of specimens of the NLR versus 53.3% (n: 8) in the LRR, showing no statistically significant difference (*p* = 0.704) (Table 2). Differences in the ER and PR hormone receptor expression of the two groups were both not statistically significant, with *p*-values of 0.236 and 0.881, respectively (Table 2).

The mean of the resection margins was comparable between the LRR (6.6 ± 4.3 mm) and NLR (6.0 ± 4.50 mm) groups, *p* = 0.543. No linear correlation was reported between the resection margin, and the recurrence timing with a Pearson correlation coefficient was 0.151.

In the LRR group, the resection margins were >2 mm in 86.6% of specimens (n: 13) compared to 77.4% (n: 141) in the NLR group, while they were < 2 mm in 13.3% (n: 2) of cases versus 23.1% (n: 42) in the respective groups, (*p* = 0.530). The margin status did not show any significant difference between LRR and NLR groups (*p*-value: 0.688) (Table 2).

The 3-year recurrence-free survival rate did not show a statistically significant difference when comparing populations with a margin width > 2 mm and <2 mm, with a relative log-rank *p*-value of 0.295 [Figure 2a].

Likewise, the difference in disease-free recurrence when comparing the margin status was not statistically significant different, with a log-rank of 0.484 (Figure 2b).

Approximately 18.2% (n: 12) of the patients of the NLR group underwent a second surgical procedure due to <2 mm margins. There were no cases of re-excision in the LRR group (*p*-value: 0.305) (Table 3).

In the LRR group, the Van Nuys score was low in 46.7% of cases (n; 7), intermediate in 53.3% of cases (n; 8), and none with a high recurrence score; differently, in the NLR group, 41.8% (n: 77) had a low score, 44.5% (n: 81) an intermediate score, and 1.1% (n: 2) a high one. The relative *p* value was 0.518 (Table 2). The 3-year disease-free recurrence, according to the Van Nuys prognostic index, was similar in the two groups (Figure 3).

Approximately 46.7% (n: 7) versus 45.6% (n: 83) of patients included in the LRR and NRL groups, respectively, did not need adjuvant hormone therapy (*p*-value: 1) (Table 3).

Within the LRR group, 33.3% (n: 5) of patients were not subjected to radiotherapy compared to 16.3% (n: 30) in the NLR one (*p*-value: 0.148). The use of an extra amount dose (boost) compared between the groups did not show any statistically significant difference (Table 3). In the NLR group, 6.1% (n: 11) received low-dose radiation therapy once a week for 5 weeks due to comorbidity and age; however, no cases of low-dose radiation were reported in the LRR group (*p*-value: 1). In the univariate logistic regression, the margin status was not a predictive factor of LRR, and the relative *p* value was 0.999 (OR: 1.142, 95% CI: 0239–5.454). A multivariate logistic regression was performed to evaluate the effect of tumor position, US findings, radiation therapy, and tumor grading on the recurrence risk. In the multivariate analysis, radiation therapy was a protective factor for LLR (Wald 5.329, *p* = 0.021, OR: 0.171, 96%CI: 0.038–0.766). A lesion site in the upper and inner quadrants resulted as a possible risk factor for LLR (Wald 4.266, *p* = 0.039, OR: 1.328, 95%CI: 1.015–1.737). The US findings, Van Nuys prognostic index, and grading were noted to provide significative predicting factors of LLR (*p* = 0.592, 1.00, 0.896, respectively) (Hosmer–Lemeshow test 0.899).

### Resection Margins < 2 mm

Out of 214 patients subjected to breast-conserving surgery, 36.9% (n: 79) presented resection margins < 2 mm. The 3-year recurrence free survival was 91.1% (Figure 1b): 18.9% (n:15) underwent re-excision with breast-conserving surgery followed by standard radiation therapy; 21.5% (n: 17) underwent mastectomy; 21.5% (n: 17) received standard radiation therapy (once a day, five days a week, for 3 weeks); 34.2% (n: 27) received a boost of radiation therapy; and 3.8% (n: 3) received low-dose radiation therapy. The three-year recurrence free survival was comparable between the groups when analyzing the treatments received by patients with unsatisfied margin criteria, and the relative log-rank was 0.091 (Figure 4a).

The age distribution was significantly different between the groups of different treatments with a relative *p* value of 0.031 (Table 4). Patients subjected to mastectomy as the second surgery presented a significantly larger lesion compared to patients treated with other strategies (*p*-value: 0.006) (Table 4). Multicentric and multifocal lesions were significative higher in patients who needed a mastectomy and standard radiation therapy (*p*-value: 0.042) (Table 4).

The expression of hormone receptors did not show any statistically significant difference, and the relative *p*-values were, respectively, 0.079 and 0.086 per estrogen and progesterone receptors (Table 4). Focusing on the histopathologic characteristics and treatment of choice for <2 mm margins, there was a significant difference in tumor grade distribution (*p*-value: 0.004) and the presence of comedonecrosis (*p*-value: 0.003) (Table 4). A higher percentage of focally positive margins was reported in patients that required surgical re-excision. The distributions of margin width after primary surgery and after re-excision surgery are both significantly different in the different treatment groups (Table 4).

The distribution of the Van Nuys prognostic index is comparable between the treatment groups (*p*- value: 0.211) (Table 4).

The recurrence rate was comparable in the five treatments group with a *p* value of 0.231. The 3-year recurrence-free survival rate is similar when analyzing the management options for <2 mm margins (log-rank 0.091) (Figure 4a).

The 3-year recurrence-free survival rate according to margin status after re-excision did not show any statistically significant difference between the treatment groups (log-rank 0.165) (Figure 4b).

## 4. Discussion

Based on the results of this single-center study, we conclude that LRR in patients with BCS is independent of margin status. The guidelines from the American Society of Surgical Oncology, Radiation Oncology, and Clinical Oncology recommend a 2 mm resection margin in patients subjected to BCS, followed by whole-breast radiation therapy [13]. The decision to use the 2 mm threshold rather than the no ink on tumor method was made on the results of a meta-analysis based on old data that are not anymore representative of the modern-day population, with weak evidence of reduction in LRR in these patients (ROR, 0.72; 95% CrI, 0.47–1.08) [13,14,15].

DCIS is not always a precursor of invasive cancer [1]. It represents 25% of breast carcinomas, and although they are mostly harmless, given the risk of progression in invasive breast carcinoma, the majority of them are still treated with surgery, often followed by radiation therapy [16,17]. Understanding and learning how to distinguish hazardous DCIS from those that are less likely to progress to invasive cancer could save many patients from overtreatment [16,17,18]. According to the current guidelines, in our institution, surgery is the first-choice treatment in all patients with a diagnosis of DCIS; exceptions are made for patients with a severe comorbidity and high anesthesiologic risk, where surveillance or palliative treatments stand as a valid option. The LORD trial highlights the psychological impact of active surveillance in patients with low-risk DCIS; however, the oncological outcome is not mentioned, and conventional treatment remains the gold standard [19,20]. Assuming that many DCIS are indolent, LRR is not necessarily associated with resection margins in DCIS. Consequently, a positive margin in indolent DCIS may also never lead to LRR due to the non-progression of the disease. Our analysis underlines that there is no correlation between LRR and margin width (Figure 2a,b). In addition, the Van Nuys prognostic index is not correlated with LRR, as shown in Figure 3. Similar results were reported in a previous analysis by Kunkiel et al. [21].

We also analyzed the sub-group of 79 patients with a resection margin < 2 mm, also considering the focal positive margins. The treatment options included were re-excision, either mastectomy or re-BCS followed by radiation therapy, and radiation therapy, considering a standard dose, boost dose, or low fractionates dose, especially in elderly patients. No difference in terms of recurrence was reported when looking at the chosen treatment (Figure 4a). Although not statistically significant, a high percentage of recurrence was observed in patients subjected to mastectomy after BCS for the management of a closer margin. This result strengthens our hypothesis that recurrence is not related to resection margins but to tumor genetics. In support of our hypothesis, our analysis reveals that patients subjected to mastectomy had a worse prognosis in terms of LRR even with resection margins greater than 2 mm. Due to the retrospective nature of the study, the choice of further treatments was based on the physician’s judgement. It is fair to think that patients who underwent mastectomy had a more aggressive disease, such multifocal microcalcifications, and/or a strong family history for BC. A more aggressive disease justifies the physician’s choice of a demolitive second surgery, and it also explains the worse outcomes. According to this, we believe that recurrence is strongly correlated to genetic factors rather than resection margins. A recent systematic review and meta-analysis considering one randomized clinical trial and eighteen observational studies showed no difference in the oncological outcomes when comparing the different options of treatments [22]. To support this result, we performed a Kaplan–Meier analysis of recurrence of this sub-group of patients and margin status, and no correlation between these two parameters was observed. Shaikh et al., in a recent analysis report, produced a similar result based on a more up-to-date sample of patients compared with a previous meta-analysis [7]. Such differences in the results could be justified by the implementation of digital and contrast enhancement mammography [23,24]. Technological advances have certainly increased the absolute number of DCIS diagnoses in the last decades, as well as of the indolent type [25]. As reported in the literature from a French group, the magnitude of overdiagnosis of DCIS associated with mammography screening remains controversial and it has led to overtreatment for many women [26]. Thanks to these newer technologies and more reliable diagnostic tools, a resection margin < 2 mm or that is focally positive could be monitored by active surveillance, avoiding unnecessary surgeries and improving the patient’s quality of life [27,28]. Although DCIS is generally considered a local disease, Hollingsworth states that DCIS could be seen as a systemic disease, given that distant metastases of DCIS can invade the ductal microvasculature of the breast without invasion. In light of those statements, DCIS could be considered as a systemic disease strongly linked to genetics [29]; however, this is out of the scope of our study, and many other studies are needed for certainty. Moreover, we think that genetic parameters could be the best predictors of recurrence and mortality.

Significant differences were reported in terms of lesion characteristics, histological parameters, and the resection margin, with types of treatments chosen during the multidisciplinary breast cancer meeting for the management of margin < 2 mm. Although it is still an open discussion, the final decision is deferred to clinical judgment, and many physicians decide for a more invasive strategy for focally positive margins or high-grade lesions, especially in younger patients [30,31]. Additionally, in our analysis, we reported a similar approach with more aggressive treatments for this type of patient.

In our analysis, age did not show a significant impact on LRR. A similar result was reported in a previous analysis by Turaka et al. [32]. This result could probably be correlated with the short follow-up of our study, which is one of the limitations of the study. Certainly, younger patients had a longer life expectancy, and the risk of LRR is higher [33,34].

Despite the more aggressive approach for the management of a margin < 2 mm or one of high grade, there was no confirmed benefit reported in terms of LRR at 3 years’ follow-up [30]. Recent studies suggest how gene expression evaluation in DCIS could help predict LRR and therefore avoid post-surgical radiotherapy for low-risk DCIS [35,36]. We strongly believe that gene assay evaluation could be the real tool to predict LRR risk and evaluate the need for re-excision of the closer margin or the need for radiation therapy. In Figure 5, we propose a potential algorithm for the management of a resection margin < 2 mm in DCIS. This algorithm could be the basis for a randomized clinical trial to evaluate the oncological safety of this proposal.

The small number of patients, the retrospective nature, and the single-center study are the limitations of our study. Another limitation was that we only used clinical and pathological factors to predict the probability of LRR and we have no data about genetic evaluations to predict recurrence. Due to those limitations, further randomized clinical trials are needed.

## 5. Conclusions

A resection margin < 2 mm seems not to be associated with LRR in spite of the patient’s age and tumor grade. Routine assessment of the gene expression of DCIS may help to predict the recurrence risk in patients with focally positive margins or <2 mm margins, therefore avoiding unnecessary surgeries and favoring radiation therapy instead. Surgical re-excision should be recommended to patients with focally positive margins and radiological evidence of the disease, and it should be always followed by active surveillance. Further randomized clinical trials are needed to evaluate the oncological safety of a conservative approach.

## Figures and Tables

**Figure 1 cancers-16-00743-f001:**
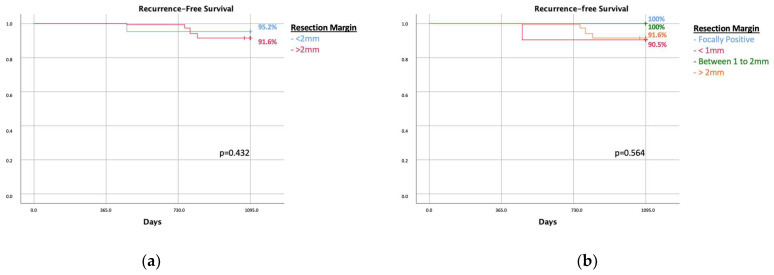
Overall recurrence-free survival in patients analyzed in the study (**a**); recurrence-free survival of 79 patients with resection margin less than 2 mm (**b**).

**Figure 2 cancers-16-00743-f002:**
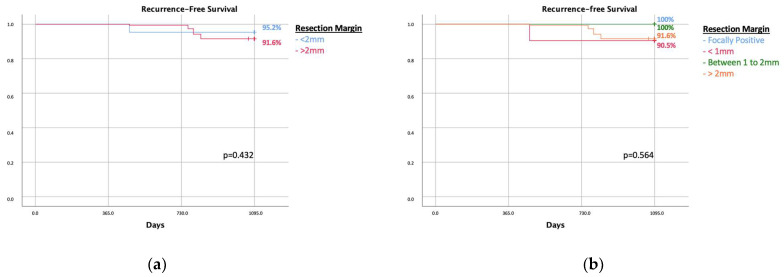
(**a**) Recurrence-free survival in patients with <2 mm margin and >2 mm. (**b**) Recurrence-free survival in patients according to margin status after eventual re-excision.

**Figure 3 cancers-16-00743-f003:**
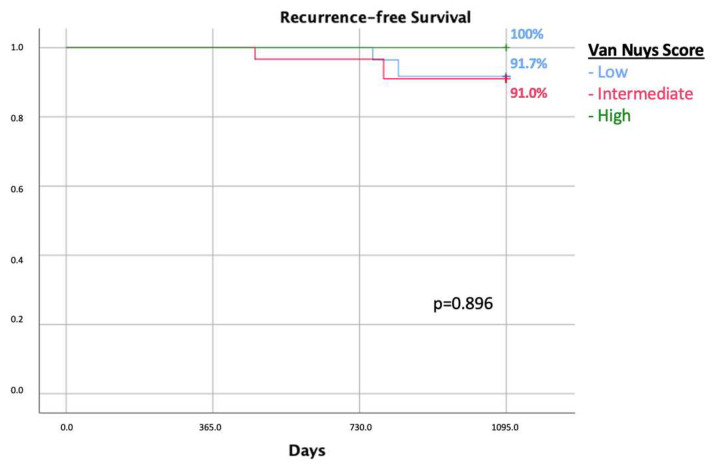
Recurrence-free survival according to the Van Nuys recurrence prognostic index.

**Figure 4 cancers-16-00743-f004:**
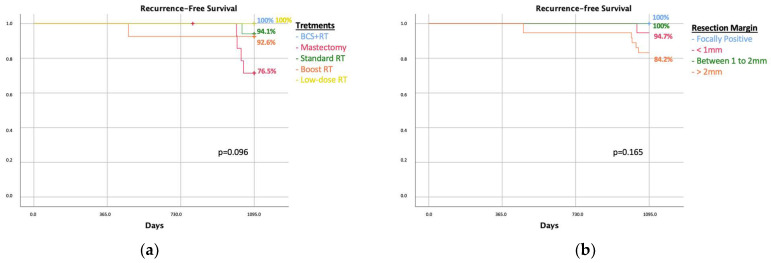
Comparison of recurrence in patients with margin less than 2 mm according to type of treatments for closer margins (**a**); recurrence-free survival of the 79 patients with resection margin less than 2 mm according to marginal status after eventual re-surgery (**b**). BCS: breast-conserving surgery, RT: radiation therapy.

**Figure 5 cancers-16-00743-f005:**
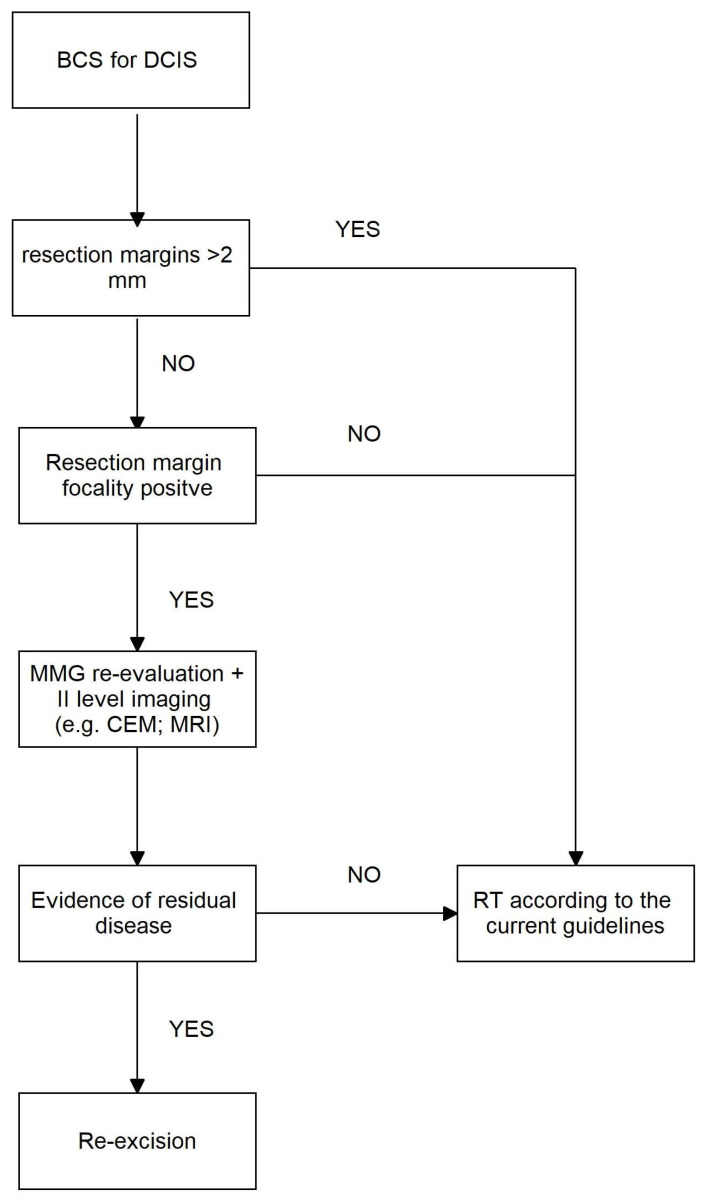
Potential algorithm for the management of resection margin < 2 mm in DCIS. BCS: breast-conserving surgery, DCIS: ductal carcinoma in situ, MMG: mammography, CEM: contrast-enhanced mammography, MRI: magnetic resonance imaging, RT: radiation therapy.

**Table 1 cancers-16-00743-t001:** Age, lesion site, characteristics, and radiological findings in the recurrence and no-recurrence groups.

	LRR Group (n = 15)	NLR Group (n = 182)	*p* Value
Age range			0.692
Age < 40 y	0	4 (2.2%)	
Age 40–60 y	9 (60.0%)	121 (66.5%)	
Age > 60 y	6 (40.0%)	57 (31.3%)	
Tumor position			0.007
Upper outer quadrant (UOQ)	0	23 (12.9%)	
Upper inner quadrant (UIQ)	4 (26.7%)	79 (44.4%)	
Lower outer quadrant (LOQ)	0	17 (9.6%)	
Lower inner quadrant (LIQ)	5 (33.3%)	7 (3.9%)	
Central portion	0	3 (1.7%)	
UOQ–LOQ	3 (20%)	3 (1.7%)	
UOQ–UIQ	3 (20%)	43 (24.2%)	
LOQ–LIQ	0	3 (1.7%)	
LOQ–LIQ	0	3 (1.7%)	
Tumor dimension range			0.377
Lesion < 15 mm	7 (46.7%)	107 (58.8%)	
Lesion > 15 mm < 40 mm	8 (53.3%)	62 (34.1%)	
Lesion > 40 mm	0	9 (4.9%)	

Values are presented as absolute numbers and percentages. We compared the values according to recurrence and no-recurrence.

**Table 2 cancers-16-00743-t002:** Histopathologic findings, margin status, and recurrence prognostic index in the recurrence and no-recurrence groups.

	LRR Group (n = 15)	NLR Group (n = 182)	*p* Value
Comedonecrosis	8 (53.3%)	81 (49.4%)	0.704
Grading	21		0.056
Low	7 (46.7%)	58 (31.3%)	
Intermediate	0	44 (24.4%)	
High	8 (53.3%)	74 (40%)	
Estrogen receptor positivity	13 (86.6%)	114 (62.6%)	0.236
Progesteron receptor positivity	12 (80%)	96 (52.7%)	0.881
Diameter (mm)	3.7 ± 3.2	4.9 ± 6.1	0.456
Closer resection margin (mm)	6.6 ± 4.2	6.0 ± 4.0	0.543
Resection margin < 2 mm	2 (13.3%)	42 (23.1%)	0.530
Margin status			0.688
Focally positive margin	0	5 (2.7%)	
Negative margin < 1 mm	2 (13.3%)	19 (10.4%)	
Margin between 1 mm and 2 mm	0	17 (9.2%)	
Margin > 2 mm	13 (86.6%)	141 (77.5%)	
Van Nuys prognostic index			0.518
Low	7 (46.7%)	77 (41.8%)	
Intermediate	8 (53.3%)	81(44.5%)	
High	0	2 (1.1%)	

Values are presented as absolute numbers and percentages. We compared the values according to recurrence and no-recurrence.

**Table 3 cancers-16-00743-t003:** Adjuvant treatments between recurrence and no-recurrence groups.

	LRR Group (n = 15)	No-Recurrence Group (n = 182)	*p* Value
Re-excision	0	12 (18.2%)	0.305
No hormone therapy	7 (46.7%)	83 (45.6%)	1.000
No radiotherapy	5 (33.3%)	30 (16.5%)	0.148
Radiotheraphy regimen			0.236
None	5 (33.3%)	30 (16.5%)	
Standard	10 (66.7%)	127 (69.8%)	
Boost	0	25 (13.7%)	

Values are presented as absolute numbers and percentages. We compared the values according to recurrence and no-recurrence.

**Table 4 cancers-16-00743-t004:** Demographics, histopathologic findings, margin status, and recurrence and prognostic index in different treatment groups for <2 mm margins.

	BCS+RT (n = 15)	Mastectomy (n = 17)	Standard RT (n = 17)	Boost RT (n = 27)	Low Dose RT (n = 3)	*p* Value
Age range						0.031
Age < 40 y	0	4 (23.5%)	0	4 (14.9%)	0	
Age 40–60 y	9 (60%)	10 (58.8%)	14 (82.4%)	17 (61.9%)	0	
Age > 60 y	6 (40%)	3 (17.6%)	3 (17.6%)	6 (22.2%)	3 (100%)	
Tumor dimension						0.006
Lesion < 15 mm	9 (60%)	4 (23.5%)	11 (64.7%)	9 (33.3%)	3 (100%)	
Lesion > 15 mm < 40 mm	6 (40%)	10 (58.8%)	3 (17.6%)	14 (51.8%)	0	
Lesion > 40 mm	0	3 (17.6%)	3 (17.6%)	0	0	
Multifocal lesion	0	3 (17.6%)	3 (17.6%)	0	0	0.042
Multicentric lesion	0	3 (17.6%)	3 (17.6%)	0	0	0.042
Microcalcifications	13 (76.5%)	13 (86.6%)	17 (100%)	21 (77.7%)	0	0.001
Comedonecrosis	12 (80%)	6 (35.3%)	14 (82.4%)	20 (74.1%)	0	0.003
Grading						0.004
Low	0	0	0	3 (11.1%)	3 (100%)	
Intermediate	0	11 (74.7%)	6 (35.3%)	8 (29.6%)	0	
High	12 (100%)	6 (35.3%)	11 (64.7%)	16 (59.3%)	0	
ER expression	3 (25%)	14 (82.4%)	11 (64.7%)	17 (63.0%)	1 (33.3%)	0.079
PR expression	3 (25%)	10 (58.8%)	11 (64.7%)	12 (44.4%)	1 (33.3%)	0.086
Resection margins						0.001
Focally positive	15 (100%)	17 (100%)	0	5 (18.5%)	0	
Margins between 0 and 1 mm	0	0	14 (82.4%)	9 (33.3%)		
Margins between 1 and 2 mm	0	0	3 (17.6%)	13 (48.2%)	3 (100%)	
Re-excision margins						0.001
Focally positive	0	0	0	5 (18.5%)	0	
Margins between 0 and 1 mm	0	0	14 (82.4%)	9 (33.3%)	0	
Margins < 1 mm	15 (100%)	17 (100%)	3 (17.6%)	13 (48.2%)	3 (100%)	
Van Nuys prognostic index						0.211
Low	15 (100%)	14(82.4%)	17 (100%)	26 (96.3%)	3 (100%)	
Intermediate	0	3 (17.6%)	0	1 (3.7%)	0	
High	0	0	0	0	0	
Recurrence	0	4 (23.5%)	1 (5.9%	2 (7.4%)	0	0.231

Values are presented as absolute numbers and percentages. We compared the values according to recurrence and no-recurrence.

## Data Availability

The data presented in this study are available upon request from the corresponding author, subject to valid justification.

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
