# Peer review of "Surgical De-Escalation for Re-Excision in Patients with a Margin Less Than 2 mm and a Diagnosis of DCIS"

_cancers, 2024, doi:10.3390/cancers16040743_

Round 1

Reviewer 1 Report

Comments and Suggestions for Authors

The de-escalation of surgery - reexcision of margins - in breast surgery, especially in DCIS is quite hot topic discussed. 2 mm margins are widely accepted in breast cancer surgery (based on na American data, NCCN guidelines). I the oncoplastic surgery era, the reexcision of the involved or narrow margins becomes a problem, so the mentioned topic is very interesting, especially to surgeons and radiation oncologist.

To change:

spelling or make carefull editing eg. : line: 59, 276, 280, 308, figure 5 RMN?

In the text the Authors use BCS (breast conserving surgery)  - well known and used abbreviation, so I really don't understand the creation and use CBS - conserving breast surgery -why and what for?

generally is it possible to make the text easier?  - different /smaller tables ?

The limitations of the study are known - single center, short follow up and small number of pts and conclusion : " ...further randomized clinical trials are needed..."  need to be highlighted.

Author Response

Dear Revieewer, thanks to your careful revision we believe that with your suggestion our manuscript in term of quality.

Best Regards 

Reviewer 1

The de-escalation of surgery - reexcision of margins - in breast surgery, especially in DCIS is quite hot topic discussed. 2 mm margins are widely accepted in breast cancer surgery (based on na American data, NCCN guidelines). I the oncoplastic surgery era, the reexcision of the involved or narrow margins becomes a problem, so the mentioned topic is very interesting, especially to surgeons and radiation oncologist.

To change:

1)spelling or make carefull editing eg. : line: 59, 276, 280, 308, figure 5 RMN?

We provide to correct all mistake evidenciated.

2)In the text the Authors use BCS (breast conserving surgery)  - well known and used abbreviation, so I really don't understand the creation and use CBS - conserving breast surgery -why and what for?

We Provide to change in BCS

3)generally, is it possible to make the text easier?  - different /smaller tables?

We provide to reduce tables and make it easier.

4)The limitations of the study are known - single center, short follow up and small number of pts and conclusion: " ...further randomized clinical trials are needed..."  need to be highlighted.

We provide to add in limitation paragraph.

Reviewer 2 Report

Comments and Suggestions for Authors

This paper describes the results of an analysis of local recurrence rate (LRR) and recurrence-free survival (RFS) based on surgical margin status (≥2 mm or <2 mm) in 197 patients with ductal carcinoma in situ in a single-center retrospective cohort study. The authors concluded that surgical margin status was not associated with LRR and RFS in these patients. My comments are as follows.

1.      A comparison of the 15 patients in the LR group with the 182 patients in the non-LR group showed no significant clinicopathologic factors. Furthermore, the second treatment modality in the 79 patients who had resection margins of less than 2 mm after BCS was not associated with patient outcome after treatment. However, patients who underwent mastectomy and had resection margins greater than 2 mm had a worse prognosis than the other treatments and resection margins, a very curious result. How do you explain this result?

2.      The authors suggest that genetic testing is the most useful method for predicting the risk of recurrence in patients with DCIS after BCS. What genetic tests are available?

3.      In considering factors associated with LR in patients with DCIS, it is important to evaluate the clinicopathologic factors of the 15 recurrent patients, detailing what treatment was given after LR and what caused their death. Did these recurrent cases progress to invasive cancer and did they have distant metastases?

4.      In terms of clinicopathologic factors in patients with recurrent DCIS, a higher recurrence rate has been reported in HER-2 positive patients What was the HER2 status?

5.      The authors showed that the status of surgical margins does not affect LRR and that LRR does not contribute to worse prognosis in patients with DCIS. Although DCIS is generally considered a local disease, some believe that DCIS is a systemic disease because distant metastases of DCIS can invade the ductal microvasculature of the breast without invasion (Narod SA, Sopik V. Breast Cancer Res Treat. 2018;169:9 -23).

6.      Finally, it is necessary to mention what is needed to avoid LR and how to improve breast cancer-specific mortality in DCIS patients.

Author Response

Dear Revieewer, thanks to your careful revision we believe that with your suggestion our manuscript in term of quality.

Best Regards 

Reviewer 2

This paper describes the results of an analysis of local recurrence rate (LRR) and recurrence-free survival (RFS) based on surgical margin status (≥2 mm or <2 mm) in 197 patients with ductal carcinoma in situ in a single-center retrospective cohort study. The authors concluded that surgical margin status was not associated with LRR and RFS in these patients. My comments are as follows.

  1. A comparison of the 15 patients in the LR group with the 182 patients in the non-LR group showed no significant clinicopathologic factors. Furthermore, the second treatment modality in the 79 patients who had resection margins of less than 2 mm after BCS was not associated with patient outcome after treatment. However, patients who underwent mastectomy and had resection margins greater than 2 mm had a worse prognosis than the other treatments and resection margins, a very curious result. How do you explain this result?

Thank you for the observations. Due to the retrospective nature of the study the choose of second treatments depends from physician. Probably patients subjected to mastectomy had a more aggressive disease: familiar history, multifocal microcalcifications. Due to possibility of more aggressive disease physician choose the mastectomy as second treatment. So according to this we believe that recurrence is correlated to genetic factors more than resection margin. We provide

  1. The authors suggest that genetic testing is the most useful method for predicting the risk of recurrence in patients with DCIS after BCS. What genetic tests are available?

For example, the Oncotype DX Breast DCIS Score® test but we preferred not to go into details because we did not perform it

  1. In considering factors associated with LR in patients with DCIS, it is important to evaluate the clinicopathologic factors of the 15 recurrent patients, detailing what treatment was given after LR and what caused their death. Did these recurrent cases progress to invasive cancer and did they have distant metastases?

Dear editor we have not cases of death or distant metastasis: probably due to the small number of cases and the short outcome. In fact, the aim of our study, due to the limitations of the retrospective nature of the study, is the recurrence and we did not evaluate progression or distant metastasis. Out of 15 cases of recurrence four cases (26.7%) presented invasive cancer and no cases of metastasis was reported with our follow-up. We provide to add in the results.

  1. In terms of clinicopathologic factors in patients with recurrent DCIS, a higher recurrence rate has been reported in HER-2 positive patients What was the HER2 status?

Dear reviewer, thanks to this important suggestion, but due retrospective nature of the study we did not have this parameter. For DCIS, in pathological report expression of HER2 was not reported. We have this data only for the 4 cases that showed invasive breast cancer as recurrence e due to small sample of cases we prefer to not consider HER2 status in statical analysis.

  1. The authors showed that the status of surgical margins does not affect LRR and that LRR does not contribute to worse prognosis in patients with DCIS. Although DCIS is generally considered a local disease, some believe that DCIS is a systemic disease because distant metastases of DCIS can invade the ductal microvasculature of the breast without invasion (Narod SA, Sopik V. Breast Cancer Res Treat. 2018;169:9 -23).

Thank you, for your suggestion we read and appreciate the manuscript suggestion and we discuss in our manuscript.

  1. Finally, it is necessary to mention what is needed to avoid LR and how to improve breast cancer-specific mortality in DCIS patients.

Thank you we provide to add in the discussion our suggestion of implement genetic test to predict as much as possible recurrence risk. But the aim of the study is a surgical de-escalation of margins re-excision in cases of resection margin <2mm. We are starting to elaborate a protocol where we will randomize patients with resection margin <2mm treated following the flowchart designed in this study.